# Study on the Relationship between Drivers’ Personal Characters and Non-Standard Traffic Signs Comprehensibility

**DOI:** 10.3390/ijerph18052678

**Published:** 2021-03-07

**Authors:** Antoni Wontorczyk, Stanislaw Gaca

**Affiliations:** 1Institute of Applied Psychology, Jagiellonian University, 30-348 Kraków, Poland; 2Department of Civil Engineering, Cracow University of Technology, 31-155 Kraków, Poland; sgaca@pk.edu.pl

**Keywords:** non-standard signs, driver personality traits, comprehensibility traffic signs, symbolic and text signs, warning and regulatory signs

## Abstract

Drivers’ incorrect perception and interpretation of the road space are among reasons for human errors. Proper road markings are elements improving perception of road space. Their effectiveness relies on traffic participants receiving the provided information correctly. The range of signs used is constantly expanding and unusual situations in traffic require use of non-standard signs or an unusual combination of existing standard signs. The aim of this study was to explore the level of comprehensibility of four different types of non-standard signs. The relationship between the level of comprehensibility of these signs and personality traits of the drivers was also studied. A total of 369 drivers were tested using a questionnaire to analyze the traffic signs comprehensibility and Five Factor Inventory (NEO-FFI). The obtained results indicate that symbolic signs, unlike symbolic and text ones, are much better comprehended by drivers. Men comprehend the significance of non-standard symbolic regulatory signs better than women. Higher level of comprehensibility of symbolic and text regulatory signs is shown by older, better educated drivers and professional drivers. The study found there is a link between personality traits of the driver and the comprehensibility of symbolic regulatory signs.

## 1. Introduction

Constantly increasing automobile mobility of societies results in various negative phenomena, such as road accidents with their severe socio-economic consequences. However, the increase in the risk of road accidents cannot be specifically attributed to growing mobility. Through numerous studies, the mechanism of traffic accidents has been better understood and this creates the possibility of effective reduction of their frequency and effects. Apart from improvements in road infrastructure, traffic management and legal regulations, special attention is paid to humans as traffic participants, as numerous studies clearly indicate that over 90% of accidents result from faulty human decisions. Incorrect perception and interpretation of the road space by drivers and pedestrians are among the reasons for human errors.

Proper road markings are one of the elements improving perception of road space and they facilitate decision-making in traffic, with the intention to effectively provide information regarding road hazards. Their effectiveness relies on traffic participants receiving the provided information correctly. This issue seems to be secondary, as the content of the signs has been regulated for many years by technical standards, and the interpretation of the meaning of the signs is a subject of driver training. However, it should be noted that the range of signs used is constantly expanding. Unusual situations in traffic may occur and these require use of non-standard signs or an unusual combination of existing standard signs. Experience also confirms that the degree of road signs comprehensibility (in the sense of supporting decision making in road traffic) varies. This also involves the diversity of personal characteristics of road users, their cognitive limitations, different experience, motivation in decision making, etc. Therefore, it is extremely important to design road signs ensuring that their understanding is not strongly affected by the aforementioned factors. This applies especially to non-standard warning signs addressing traffic safety hazards. 

Unusual signs may appear in situations where the use of standard solutions proves to be ineffective or when new, sometimes surprising solutions for road infrastructure users are implemented. Due to the form of road infrastructure, new solutions often require additional unusual markings (e.g., showing how the infrastructure is used) in the initial period of their use, and these markings are specific to each site. Developing intelligent traffic management and control systems also requires an extension of the range of standard road signs currently used. 

The increasing use of traffic signs with non-standard content requires answering at least the following general questions:What might be the effectiveness of non-standard traffic signs in providing additional information warning about traffic safety hazards?What should be the form of non-standard traffic signs to make their level of comprehension and impact on the decisions of traffic participants as high as possible?

The study undertaken by the authors is an attempt to answer the above questions with a focus on one of the aspects of assessing the comprehension of non-standard signs, i.e., the impact of drivers’ personality on the comprehension of these signs. The study aims to investigate the reception conditions of non-standard traffic signs and decision- making process based on them. A hypothesis, stating that the impact of socio-demographic and personality features on the level of comprehension of non-standard traffic signs depends on the form and function of these signs, was formulated. In practical terms, it may be an indication to restrict the use of some forms of non-standard signs. The application of signals, whose understanding depends largely on the personal traits of the driver, should be avoided. 

## 2. The Study of Traffic Signs Comprehension 

Due to the role of traffic signs as means to influence the road users’ behavior, the problem of the comprehension of traffic signs content is a subject of numerous scientific studies and expert assessments. The level of traffic signs comprehensibility and its susceptibility to the following factors was studied: sign ergonomics [1,2]; text and symbolic messages combination [3]; sign coloring [4]. Most studies concentrate on the processing of the information which drivers receive from traffic signs. They were analyzed in terms of the amount to process [5,6,7], information placement [8,9,10] and its configuration [11]. An important part of the study, due to practical aspects, is the assessment of drivers’ comprehensibility of traffic signs in various conditions [3,12] and impact of their placement, e.g., curves, on the drivers’ behavior [13,14,15].

Traffic signs comprehensibility is related to the drivers’ psychosocial features, such as gender, age and driving experience [13,14,15,16,17,18,19,20,21,22,23,24]. A relationship was also found between the level of road signs comprehensibility and the drivers’ age groups, including their health status and levels of cognitive impairment [25]. An important group of psychosocial factors affecting the traffic signs comprehensibility are drivers’ cultural factors (novice, students, tourists or elderly drivers with cognitive problems) [20].

In the process of road signs content reception by drivers, the so-called distractors can occur. These are situations in which traffic signs are accompanied by commercial advertisements [26,27,28] or they even contain commercial content [29]. 

Previous studies have also found that the amount of information carried by the signs has a significant impact on their reception effectiveness [5,22]. If the number of messages is excessive, their reception decreases. Similarly, placing signs along busy routes reduces their reception by up to 25% compared to standard reception [30]. 

Studies on the comprehensibility of traffic signs also took into account psychological factors embedded in driving style or the need for cognitive closure [24].

Personality traits affect not only the reception of the road signs content, but are also among explanatory variables in the models used for traffic safety measures prediction. This was confirmed by studies from both the early period of automotive development [31,32,33,34], as well as the later ones [35,36,37,38,39,40,41,42,43,44,45,46]. Modern studies also lead to similar conclusions [47,48,49,50,51,52,53,54,55,56]. 

Personality as a human disposition trait has many aspects, however, in general it can be described as relatively stable (over time) behavioral tendencies of people, or as mental traits which lead to certain behavior. Many concepts and models of personality traits have been developed in psychology but factor model known as the Big Five has gained the most popularity [57]. It describes the five dimensions of personality: openness, agreeableness, conscientiousness, neuroticism and extraversion. 

Studies on drivers’ behavior in traffic have found several patterns [44,51]. Extrovert drivers tend to be less vigilant and take greater risks. People with high levels of neuroticism are easily distracted, less often attempt to control the environment. Conscientiousness is positively related to safe driving, because it is associated with the ability to plan, self-control and correct decision-making. Lack of agreeableness is associated with accidents through the mechanism of aggression in terms of emotions and behavior. Finally, openness is positively associated with a tendency to accidents, because drivers with high scores in this scale tend to prefer routine driving, experimenting and improvisation.

The data collected within the framework Experimental road markings in terms of road users behavior (Project DZP/RID-I-36/5/2016 supported by the National Centre for Research and Development and the General Director for National Roads and Motorways) were used in the present behavioral study on non-standard traffic signs. Three main objectives of the study were formulated:to check whether the level of comprehensibility is influenced by the functional group of signs, including whether symbolic signs are better comprehended than symbolic-text ones,to determine whether a higher frequency of encountering them is related to their level of comprehensibility,to confirm or exclude a relationship between the personality traits of drivers and the level of comprehension of non-standard signs. This involves an attempt to isolate personal characteristics or profiles of several personal characteristics that promote better comprehension of non-standard signs.

## 3. Research Methodology 

The scope and method of research refer to pilot implementations of non-standard traffic signs in Poland. In 2017–2018, the General Directorate for National Roads and Motorways placed non-standard signs, hitherto unknown to drivers, on selected sections of the motorways and sections of expressways. Their purpose was to improve safety and improve traffic. It was decided to check these non-standard signs in terms of their understanding by vehicle drivers. 

Two types of traffic signs with content different from typical traffic signs, as described in the Polish traffic law, were selected for the study: (a) regulatory signs and (b) warning signs. Regulators and signs are important due to decisions made by vehicle drivers—following these signs is required or the content of the sign is necessary to choose the right path. Warning signs provide warnings that are helpful for decision-making by vehicle drivers but may be ignored by them. Both groups of tested traffic signs were brand new types on Polish roads. The two groups of tested traffic signs include symbolic and combined symbolic-text signs. Finally, four different types of signs were used for the experiment (Figure 1):

According to the American National Standard Institute (ANSI Z535.3), 85% correct response criterion was adopted for confirmation of traffic signs’ comprehensibility and familiarity [58]. In this study, tested signs were completely unknown to Polish drivers, they were not the subject of drivers’ training and no campaigns regarding their importance or meaning were made, therefore we adopted soft comprehensibility criteria. A level of 60–85% of signs comprehensibility was classified as average. Lowering the threshold of comprehensibility to 60% was also due to the fact that the studied signs were used in practice and an “educational effect”, which would increase level of their comprehensibility over the level declared in the surveys, could be expected. This assumption was accepted as an important criterion for leaving the signs used on the road for further study. Signs comprehensibility below 60% was assumed as poor. In such cases, further analyses were undertaken to determine whether the sign should remain or be removed from the road. These analyzes were not the subject of this study.

In addition to graphic material in the form of signs, two part questionnaire was also used. The first part covered the respondent’s descriptive data, such as: age, gender, Driving License Category, driving experience, education. The second part investigated traffic signs familiarity and comprehensibility. Traffic signs familiarity was measured by asking the following question: “Have you ever encountered such a sign on the road while driving?” When answering, the respondents had a choice of five possible variants of the answer: 1. “No, never”, 2. “I think so, but I don’t remember where”, 3. “Yes, several times on different roads”, 4. “Yes, on the highway”, 5. “Yes, abroad”. Choosing the answer variants 3, 4 or 5 (or all three together) indicated that the examined person had met such a sign on the roads, and therefore knows it. If the tested person chose variants 1 or 2, we decided that the tested sign was unknown to him.

Comprehension was measured separately for each sign. In the questionnaire, under the graphic image of the sign, there was a list of different answer variants (from 5 to 9). The list of answer options for each character was selected using the Competent Judges method. From the list of proposed variants of answers describing the meaning of a given symbol, only one was correct. If the examined person chose the correct variant of the answer, it was assumed that they understood the meaning of this sign. The number of variants of the proposed answers to the meaning of a given sign depended on its complexity. For example, if the sign contained only two or three elements (symbol and text), there were fewer (5–6) variants of the proposed answers. If the sign was more complex (a few symbols and a text), the number of answer variants was greater and ranged from 7 to 9. If the respondent found that all the proposed variants of the answers describing the meaning of the sign were incorrect, they could provide their own understanding of this sign.

To measure drivers’ personality traits, a commonly used psychology tool—Five Factor Inventory (NEO-FFI) by Costa and McCrae in Polish adaptation [59] was used. The Five Factor Inventory is broadly used in psychological studies to measure the personality traits. It was decided to apply this method also in these studies. This tool measures Big Five personality traits, namely: neuroticism, extraversion, openness to experience, agreeableness, and conscientiousness. It consists of 60 items, 12 questions with a five point scale. Cronbach’s alphas were 0.69 for neuroticism, 0.89 for extraversion, 0.62 for openness to experience, 0.78 for agreeableness, and 0.87 for conscientiousness.

Table 1 presents the survey sample distribution according to drivers’ characteristics.

The study was conducted in 2017–2018 in south-eastern Poland, i.e., the following provinces: Lubelskie, Podkarpackie, Małopolskie, Świętokrzyskie.

The test persons were selected randomly. In the case of professional drivers (Lorry Driver, Bus, Driver, Driving Instructor and Taxi Driver), the random selection was made on the basis of data obtained from companies employing such drivers. Then a trained person made an appointment with them and, having obtained their consent, conducted the research. In turn, amateur drivers were selected for tests in natural conditions. Firstly, out of several hundred petrol stations in four voivodships of southern Poland, four were randomly selected—one in each voivodship. Then, at the stations selected for research, the trained person interviewed the drivers stopping at the station.

After obtaining informed consent from the tested drivers, they were showed a traffic sign and asked to look closely at it and indicate familiarity. Next, comprehensibility of a given sign was measured—the subjects were asked to read the list and select one answer from several, which in their opinion best explained the meaning of the sign. The time was not limited and the subjects were allowed to return to tasks already completed if they considered that they had given the wrong answer. A total of 369 drivers were tested. Data regarding psycho-demographic variables is summarized in Table 1. 

The data presented in this paper was analyzed using statistical software Statistical Package for the Social Sciences (SPSS), version 22.0 (SPSS Inc., Chicago, IL, USA). Statistical analysis included: descriptive analysis, independent-samples t-test, one-way analysis of variance (ANOVA), and post-hoc multiple comparisons.

## 4. Results

The research results are described below, starting from familiarity and comprehensibility of traffic signs in four selected groups:(a)Symbolic Regulatory Signs (SR),(b)Symbolic and Text Regulatory Signs (STR),(c)Symbolic Warning Signs (SW),(d)Symbolic and Text Warning Signs (STW).

Then the relation of personal characteristics of drivers (i.e., age, gender, license category, educational level, driving experience, monthly income, and the number of traffic violations during the last five years) with the familiarity and comprehensibility of selected groups of traffic signs were assessed.

### 4.1. Traffic Signs Familiarity and Comprehensibility

General data on the familiarity and comprehensibility of both individual traffic signs and four groups of traffic signs separated in studies are presented in Table 2.

The tested drivers showed an average level of comprehensibility only in the case of three traffic signs. That is one symbolic regulatory sign (SR2) and two (SWS1 and SWS2) from the Symbolic Warning Signs group. In case of the remaining signs, the comprehensibility level by the subjects was below average or very poor. There was no case of high comprehensibility (defined as above 85% of the responses) of the content of a sign both in individual signs and the studied four groups. The result on the borderline of average comprehensibility (above 60%) was obtained only for the group of Symbolic Warning Signs.

As the result of comparative study of the level of comprehensibility between different groups of signs, the drivers showed better comprehensibility in the case of symbolic signs (Table 2). The highest level of comprehensibility of traffic signs was found in case of symbolic warning signs and symbolic regulatory signs. A much poorer comprehensibility was observed in the case of symbolic-text regulatory and warning signs. The differences in signs comprehensibility turned off to be statistically significant (Table 3).

In order to estimate statistical significance of differences understanding when analysing 4 separate groups of non-standard signs, the mean values of the correctness of understanding the meaning of a given group of signs by each of the respondents have been found. 1 point was given for a correct recognition of each sign (and not a group of signs). If the recognition was incorrect, then a number 0 was assigned. So, the minimal number of points was 0 (in the case when none of signs was correctly described), while the maximum was: 5, 4 and 3 for SR/STR, SW and STW signs, respectively. Having the information about the mean values for each group of signs (369 checked respondents), the statistical significance of differences between them was possible to assess. 

Repeated ANOVA analyses confirmed that the differences between the averages obtained in the individual groups of the analyzed traffic signs were statistically significant (F (4.957) = 222.78; *p* = 0.001). (N—sample size; SD—standard deviation; F-distribution; *p*-value for statistical significance)

The assessment of the declared level of comprehensibility of the sign should include signs familiarity. Frequent encountering of a non-standard sign on the way can stimulate cognitive curiosity which can lead to its better comprehensibility. In case of examined group of signs, the level of familiarity of the signs can be assessed as average only in the case of two signs (SW.1 and SW.2) from the Symbolic Warning Signs group. In the remaining cases, the respondents showed a varied but below average or very low level of familiarity of the signs assessed. The relationship between the level of familiarity of the signs from a given group and the level of their comprehensibility was assessed by determining the regression relationship in the examined sample of people.

A very low value of R^2^ determination coefficient was obtained in the analyzed relationship for the entire sample of the signs studied (Table 4). Considering individual groups of non-standard signs, the impact of the level of their familiarity on the level of their declared comprehensibility varied. A higher level of familiarity of Symbolic and Text Regulatory Signs promoted their better comprehensibility. A similar relationship was also found for Symbolic Warning Signs (Table 4).

The analyses described above confirmed that the non-standard signs tested did not show the desired level of comprehensibility. However, the selected groups of symbolic signs with an average level of comprehensibility were observed. The hypothesis that familiarity of a sign (encountering it on the road) can significantly affect the level of its comprehensibility has not been fully confirmed. Such a relationship has been demonstrated only in relation to Symbolic Text Regulatory Signs and Symbolic Warning Signs.

### 4.2. Effect of Driver’s Gender on Traffic Signs Comprehensibility

A statistically significant relationship was found only for Symbolic Regulatory Signs *t* = 7.465, *p* < 0.007. For this group of signs, men show a higher level of comprehensibility (Table 5). 

### 4.3. Effect of Drivers’ Age, Driving Experience, Educational Level and License Category on Traffic Sign Familiarity and Comprehensibility

In the next step, the presence of statistically significant relationships between socio-demographic variables (i.e., age, driving experience, type of driving license category and educational level) and traffic signs familiarity and comprehensibility were investigated. The results of statistical analyses are summarized in Table 6.

It should be noted that the evaluated level of familiarity and comprehensibility of non-standard traffic signs refers to the signs which are not commonly found on Polish roads. Some of the tested drivers may have encountered non-standard signs on some sections of national roads in Poland (experimental applications). The surveyed drivers could also come across similar traffic signs on the roads abroad. Therefore, the responses of the surveyed drivers about the signs familiarity also contain an element of interpretation and associations with other signs, which can also be affected by their socio-demographic characteristics.

In case of the variable “age of the surveyed person”, relationships with the declared familiarity of signs were detected only for the Symbolic text regulatory signs (F (4.364) = 2.354 *p* < 0.041) and Symbolic Warning Signs (F (4.364) = 2.534 *p* < 0.021). In these two cases, younger drivers showed a higher level of declared familiarity of non-standard signs. In contrast, the statistically significant impact of the variable “age of the surveyed person” on the level of signs comprehensibility was detected only in case of Symbolic regulatory signs (F (4.364) = 3.951 *p* < 0.047). Younger drivers showed a higher level of comprehensibility of Symbolic regulatory signs.

In case of the variable “experience of the surveyed person” significant statistical impact on the declared familiarity of signs was detected in three groups of signs: (a)Symbolic Regulatory Signs (F (5.363) = 27.72 *p* < 0.000),(b)Symbolic and Text Regulatory Signs (F (5.363) = 2.511 *p* < 0.023),(c)Symbolic Warning Signs (F (5.363) = 7.378 *p* < 0.000).

As expected, as the experience increases (the number of kilometers driven per year), the level of familiarity of non-standard signs also increases.

However, it is surprising that there is no statistically significant impact of experience on the level of comprehensibility of non-standard signs in all distinguished groups of signs.

In case of the variable “driving license category”, it turned out that it affects the familiarity of non-standard signs of the following groups:(a)Symbolic Regulatory Signs (F (5.363) = 7.462 *p* < 0.006)(b)Symbolic and Text Warning Signs (F (5.363) = 3.595 *p* < 0.007).

Truck and emergency vehicle drivers showed a higher level of familiarity of non-standard signs. However, the impact of the variable “driving license category” on the level of signs comprehensibility turned out to be statistically significant only in case of symbolic-text regulatory signs (F (4.365) = 2,855 *p* < 0.023). A higher level of comprehensibility (as well as familiarity) of this group of signs was demonstrated by truck and emergency vehicle drivers.

No statistically significant relationship was found between the level of education of the subjects and the familiarity and comprehensibility of non-standard signs.

To sum up this part of the study, it can be stated that the declared level of comprehensibility of non-standard signs depends on their form and meaning in terms of the impact on traffic. Some socio-demographic variables affect the level of this comprehensibility, but this impact varies depending on the form and meaning of non-standard signs. The study has not confirmed the hypothesis that the drivers’ experience can positively affect the correct comprehensibility of non-standard signs. The impact of the “driving license category” on comprehensibility of non-standard signs was also limited. 

### 4.4. The Effect of Driving Personality Traits and Traffic Signs Comprehensibility

The second important aim of the study was to detect relationships between the driver’s personality traits and the level of comprehensibility of signs from a given group. It was assumed that possession of a given personality trait or a specific constellation of several traits with a certain intensity (a profile of personal traits) may affect the level of comprehensibility of non-standard signs. For this purpose, cluster analysis was performed for each of the four separate groups of non-standard signs. In the cluster analysis, a link was sought between the declared level of sign comprehensibility and variables describing the driver’s Five Factor Model of personality. As a result of the analyses, typical personality profiles of drivers indicating a better or worse comprehensibility of non-standard signs were identified. The analyzes were conducted separately for separate groups of signs.

Drivers’ personal profiles in terms of comprehensibility of the individual groups of signs were established on the basis of the analysis of the results obtained in the Big Five test [57] and the signs comprehensibility (Figure 2, Table 7).

In the case of Symbolic regulatory sings (SR), cluster analysis indicated three personality profiles that revealed statistically significant relationship with the comprehensibility of the signs studied (Figure 2—SR).

The highest level of comprehensibility of symbolic signs regulating traffic was demonstrated by drivers with an average intensity of: extraversion, agreeableness and openness to experience, low level of neuroticism and high conscientiousness (profile cluster III). The other two profiles (clusters 1 and 2) were drivers who have shown a poor comprehension of non-standard symbolic regulatory signs, however, differing in personality traits. Profile cluster 1 is a driver with high results in all five personality traits (extraversion, agreeableness, openness to experience and conscientiousness). Profile cluster 1 is a driver representing people with average intensity of personality traits: agreeableness, openness to experience and neuroticism as well as low levels of extraversion and conscientiousness.

In case of comprehensibility of Symbolic text regulatory signs (STR), cluster analysis has identified two statistically significant and clearly different personal profiles (Figure 2—STR). Drivers who comprehend this group of signs better are characterized by average results in all five personality traits (profile cluster 2). On the other hand, drivers with features classified as above average in five personality traits show a worse comprehension of these signs (Profile cluster 1).

In case of comprehensibility of the meaning of Symbolic warning signs (SW) and its connection with personal traits, cluster analysis (Figure 2—SW) led to results very similar to those of Symbolic text regulatory signs.

Drivers with below-average results in all five personality factors (profile cluster 2) show a better comprehensibility of these signs than drivers with above-average results (profile cluster 2).

In case of Symbolic and text warning sings (STW), no statistically significant relationship was found between the drivers’ personality traits and the comprehensibility of this group of signs. Although cluster analysis has identified two different profiles of drivers’ personality traits in terms of comprehensibility of these signs (Figure 2—STW), it did not show statistically significant differences between them in terms of the level of their correct comprehensibility.

## 5. Discussion of Research Results

Obtained research results clearly indicate that symbolic signs, unlike symbolic and text ones, are much better comprehended by drivers. The understanding of symbolic signs that improve traffic, symbolic regulatory sings, and symbolic warning signs, was comparable. However, a clearly weaker level of comprehensibility was observed in case of symbolic and text signs, in particular Symbolic and Text Warning Signs, by which designers intend to increase the level of traffic safety. However, it should be noted here that the tested Symbolic and Text Warning Signs were completely new signs, just being prepared for use as a part of a pilot project.

Generally, the results of the described studies are consistent with those obtained in most studies of other authors [3,13,21,22]. The exception are results of study [3], in which it was found that adding text to a symbolic sign significantly improves its comprehensibility by drivers. In our study, the opposite effect was observed: symbolic and text signs were much less comprehended than signs without text, containing only symbols. In comparison of two groups of symbolic and text signs, i.e., Symbolic and Text Regulatory Signs and Symbolic and Text Warning Signs, more favorable results of sign comprehensibility studies were obtained in case of Symbolic and Text Regulatory Signs. This unexpected result is influenced by better familiarity of this group of signs. Several signs from this group (STR 1, STR2 and STR 4) have been more widely used on Polish roads (as experimental applications) and drivers have probably already learned to read their meaning, despite the complex content presented on these signs.

A strong argument for the fact that non-standard symbolic signs are better comprehended by drivers than symbolic and text ones is the lack of connection between the level of comprehensibility of these signs and their familiarity (encountering on the road). This observation should be sought for two reasons. Firstly, the authors used unusual (experimental) signs which drivers, with a few exceptions, did not come across on the road. Secondly, most of the symbolic-text signs contained too many messages (both textual and symbolic) which makes them difficult to integrate by human attention resources in a short time. It is well known that these resources are limited [60,61]. Some symbolic signs were also very poorly comprehended by drivers (SR4, SR5), their content was conveyed using unusual symbols, incompatible with the context of meaning. This could have had a negative impact on their comprehension. Jamson and Mrozek [19] have already observed this problem, pointing to the need to standardize traffic signs in terms of color, shape and their physical and spatial features. It seems that it is also important to take into account the meaning of the symbol itself (shape, color, message model) in the designing process. A low amount of textual symbols and messages (STW1, STW2, STW3, STW4) could also have affected the low level of drivers’ comprehension of non-standard symbolic-text signs. Other researchers also draw attention to this problem in their work [5,22,30]. Even if the symbols on the traffic sign are consistent with the context, due to their abundance, it is difficult to fully integrate their meaning by human attention resources, as these select messages by choosing easier or better comprehended ones, bypassing the more difficult or even ignoring them in the process of receiving and processing information. This phenomenon is particularly visible in drivers with worse cognitive resources and in old age. Such a finding is consistent with modern studies on the process of central processing of cognitive reserves [60,61,62,63,64,65].

Analyzing the problem of non-standard traffic signs comprehensibility, the authors extended their study to look for possible socio-demographic factors. Two interesting results were obtained in these studies. First of all, it was found that men comprehend the significance of non-standard symbolic regulatory signs better than women. However, no such differences were found in relation to the other groups of non-standard signs analyzed. The result of our study is consistent with several other studies [16,17,19].

The second important observation concerns the different level of comprehensibility of non-standard signs depending on age, education and driver license category. It was found that such diversity applies only to symbolic and text regulatory signs (i.e., more difficult from the point of view of their comprehension). Higher level of comprehensibility of this group of signs is shown by older drivers with a higher education and professional qualifications to drive a vehicle (bus and lorry drivers, driving instructors). This result should be interpreted as the result of both greater experience (older drivers) and having better resources and cognitive reserves (higher education) [63,64]. Greater social responsibility (professional drivers, lorries, transporting people) and a high sense of competence (driving instructors) are also important. Slightly more complex relationships were found between some socio-demographic features and familiarity of signs from four groups. This applies in particular to symbolic regulatory signs, as well as symbolic text regulatory signs. Their better familiarity was observed for younger drivers than older ones, with extensive experience in driving, as well as among professional drivers (truck drivers, driving instructors and emergency vehicles drivers).

Interesting results were obtained when analyzing the connection between traffic signs comprehensibility and personality traits of the respondents. Particularly noteworthy are two statistically significant relationships, i.e., linking the personality traits of the driver with the comprehensibility of symbolic regulatory signs. In the described studies, a generally better comprehensibility of the meaning of non-standard symbolic signs than symbolic-text ones was found. In case of symbolic regulatory signs, it turned out that it is possible to indicate a profile of personal traits, which even in a model way can explain a better comprehensibility of the meaning of these signs. Such a profile corresponds to the average level of extraversion, agreeableness, openness to experience and neuroticism, as well as above-average conscientiousness. Many studies on the behavior of drivers have indicated that a high level of conscientiousness is associated with safe driving, better planning, self-control and making correct decisions [44,51,53,54,55,56]. In turn, high extraversion weakens vigilance, which interferes with the correct comprehensibility of the meaning of signs. High neuroticism causes that drivers observing the surroundings of the road often get distracted losing control over following signs along the road. 

The personal traits linked to poor signs comprehension was presented by drivers with a profile that corresponds to high results in all five personality factors (extraversion, agreeableness, openness to experience and conscientiousness). The explanation of these relationships is possible from two perspectives. Driving a vehicle is a complex activity which involves several processes: sensory, cognitive, decision-making and personal. Many studies have indicated that personality traits are an important factor in risky behavior among drivers. It was indicated that drivers were neurotic drivers [66] deliberately break the road traffic rules (intentional road traffic rules violation) and are involved in more accidents [67,68,69]. Neurotic and extraverted drivers are prone to aggressive behavior in road traffic [66,70,71]. Therefore, from this perspective, the results obtained by us are consistent with these studies. Neurotic and extroverted drivers understand the meaning of signs worse because they are generally worse at processing data, need more time to understand them, but also make more conscious errors, because these processes are accompanied by severe anxiety and impulsiveness. 

From the second perspective, personality traits such as agreeableness [66,72,73,74] and conscientiousness [75,76] were previously indicated as features favoring positive behavior in road traffic. In our study, high scores on the agreeableness and conscientiousness scale turned out not to be favorable to the understanding of non-standard road signs. However, this contradiction is only apparent. Understanding the meaning of road signs is also related to the efficiency of sensory, cognitive and decision-making processes. Strong motivation to perform a correctly complex cognitive task limits these resources [77]. As a result of this process, the performance itself deteriorates. The process of limitation of peripherical eye scanning [78] and an impairment in the perception of relevant stimuli [79] follows. Strong motivation to perform the task correctly reduces the overall efficiency of cognitive resources [80] and even leads to longer drivers’ reaction times [77]. Among people with the predominant features of conscientiousness and agreeableness, in a situation where they need to perform a task correctly, the experience of stress increases [81]. In addition, the situation is worsened by the time deficit in which this task must be performed. Some studies have emphasized that conscientiousness and agreeableness are associated with a pro-social attitude and achievement motivation [82,83,84,85]. It is also worth emphasizing that in the human population it is rare for individuals to have all five personality factors at their maximum intensity. Usually they are arranged in different profiles. For example, when some features dominate (extraversion, neuroticism) others are weaker (conscientiousness and agreeableness) or vice versa.

This unfavorable profile of drivers’ personal traits from the point of view of the correct comprehensibility of non-standard traffic signs, could be compensated by learning their meaning beforehand or by repeatedly experiencing their presence in traffic, although (in the latter case) this may be an unfavorable strategy from a safety point of view. 

Obtained results contain important information regarding the conditions for introducing non-standard signs in practice. It should be noted that the percentage of people with personal traits corresponding to profiles for which a low level of comprehension of non-standard signs is obtained is significant in the adult population, including drivers. Therefore, the use of non-standard signs requires caution.

Results of the presented research are limited by the small statistical sample of professional drivers participating in the research, namely: Bus Drivers, Taxi Drivers and Driving instructors. In the research, people from these groups were selected by randomization. In the future research the number of drivers from these three groups should be increased because it turned out to show a higher knowledge of non-standard signs compared to the other drivers. However, preliminary analyses show that one should not expect important effect of the increased ratio of experienced drivers in the investigated sample on the statistical analysis similar to this being the core of recent studies.

One should remember that the reaction study was not carried out in realistic conditions. Because of this, responders had significantly longer time for reaction compared to real situations. Therefore, more significant attention has been paid to the assessment of analysis differences in each separated groups than to the level of each signs understanding.

As a future work, tests in a driving simulator are planned, which should eliminate the specified limitations.

## 6. Conclusions—Practical Recommendations

The conducted research on the problem of understanding non-standard signs by drivers confirmed the regularity observed in other studies, i.e., better understanding of non-standard symbolic signs [3,13,21,22] as opposed to symbolic-text characters. A better understanding of the symbolic signs is shown by men rather than women, younger drivers, drivers with more driving experience, truck drivers and emergency vehicle drivers. However, it is worth remembering that placing several symbols on road signs significantly worsens their understanding, as our own research showed. Addition of text to symbols on non-standard signs did not improve their understanding, and this was observed also in other studies [3]. Most likely it is related to the number of symbols and words placed on one non-standard sign, which should be limited. Moreover, the symbols and text placed on the signs must be legible and understandable for drivers. Symbols should refer to commonly recognizable pictograms, and the text must be short, communicative and unambiguous.

Significant influence of drivers’ personal traits on understanding the meaning of non-standard signs was also found. Important for the understanding of non-standard signs are features commonly considered to be associated with both dangerous behaviour in road traffic (including neuroticism and extraversion) and also to those commonly recognized as favouring correct behaviour in road traffic (such as conscientiousness and agreeableness). The first group of personal traits is not conducive to careful tracking of road signs and effective scanning of the content of these signs (symbols and text) due to impulsive behaviour of drivers. On the other hand, the second group of drivers with conscientiousness and agreeableness as predominant personal traits, may experience stress in a situation of not understanding the meaning of non-standard road signs. Strong stress limits the efficiency of sensory and cognitive resources, which leads to impaired performance in road traffic.

The described conditions of drivers’ comprehensibility of the meaning of non-standard traffic signs indicate the need to use appropriate procedures to implement them to minimize the risk of ineffectiveness of this type of signs. Particular attention should be paid to behavioral studies in these procedures, including tests of non-standard signs comprehensibility before their practical introduction.

Based on our study and studies of other authors, the following general recommendations can be made regarding the practical implementation of non-standard traffic signs:non-standard signs should be used only in cases where standard solutions prove to be insufficient or there are no standard signs addressing the need to influence the traffic in the presence of specific road situations;the form and content of non-standard signs should meet the basic ergonomic requirements applicable to standard signs. Excess content and its unfavorable distribution makes it difficult to properly receive information also in the case of non-standard signs. The unusual form of the sign may draw attention to it, but it is not a guarantee that one will comprehend its informational message;practical application of a non-standard sign in a specific place should be preceded by surveys of its comprehensibility. The result of such study may indicate the need for corrections of the form and content of the sign or its rejection;despite the generally higher level of comprehensibility of symbolic signs, the use of non-standard symbols on traffic signs should be preceded by an educational action. The study confirms the risk of a low level of comprehensibility of some symbols, despite their familiarity (i.e., encountering non-standard signs placed along roads);receipt of the information of some non-standard signs may be affected by local conditions of the place of use. Therefore, pilot applications of non-standard signs should be monitored to identify their actual impact on the behavior of road users.

## Figures and Tables

**Figure 1 ijerph-18-02678-f001:**
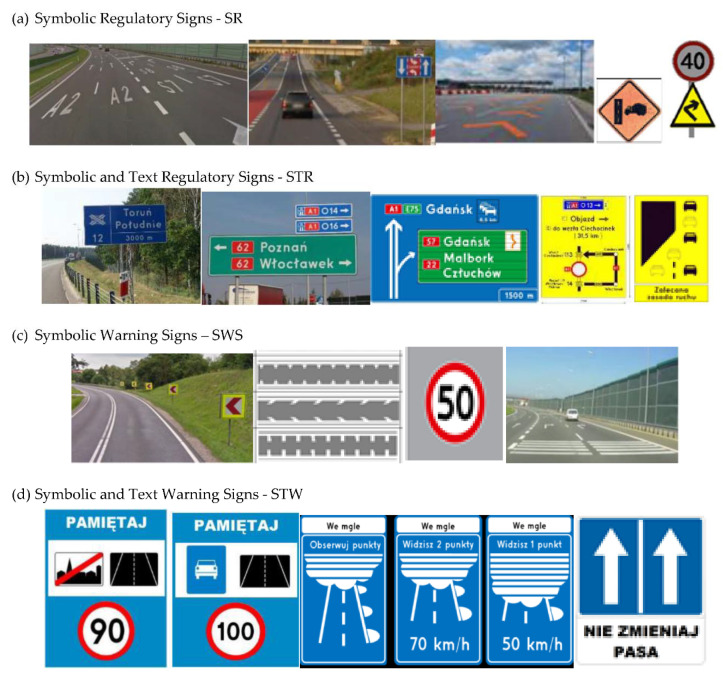
Studied regulatory and warning signs.

**Figure 2 ijerph-18-02678-f002:**
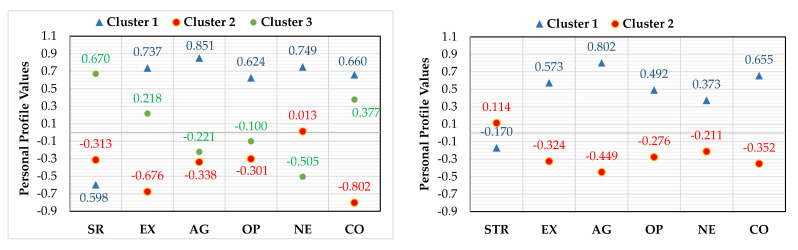
Personal profiles of drivers regarding comprehensibility.

**Table 1 ijerph-18-02678-t001:** Distribution of the survey sample according to driver’s characteristics.

Driver’s Characteristics	Categories	Frequency	Percent
Gender	Male	218	59
	Female	151	41
	Total	369	100
Age group	18–25 years	86	23
	26–35 years	104	32
	36–45 years	60	14
	46–55 years	72	19
	56–65 years	47	12
	Total	369	100
Driving Status	Amateur Driver	272	74
	Lorry Driver	47	12
	Bus Driver	26	8
	Driving instructor	8	2
	Taxi Driver	16	4
	Total	369	100
Driving Experience	1–5000 km per year	51	14
	6–10,000 km per year	64	16
	11–15,000 km per year	51	14
	16–20,000 km per year	40	11
	21–30,000 km per year	34	9
	31–40,000 km per year	20	6
	>41,000 km per year	109	30
	Total	369	100
	Bachelor degree	100	28
Educational level	High school	115	31
	MS/Ph.D.	154	41

**Table 2 ijerph-18-02678-t002:** Comprehensibility and familiarity of different types of signs.

**Symbolic Regulatory Signs—SR**			**Symbolic and Text Regulatory Signs—STR**		
**Signs**	**Comprehensibility**	**Familiarity**	**Signs**	**Comprehensibility**	**Familiarity**
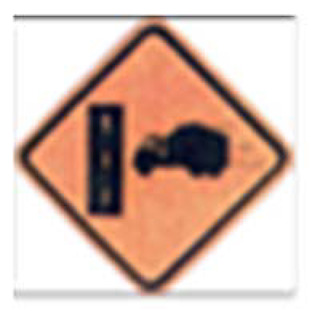	SR1 54%	5%	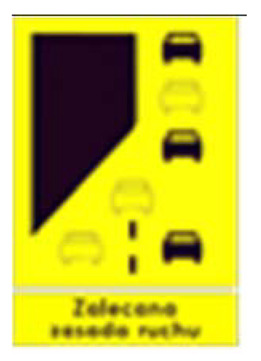	STR1 35%	56%
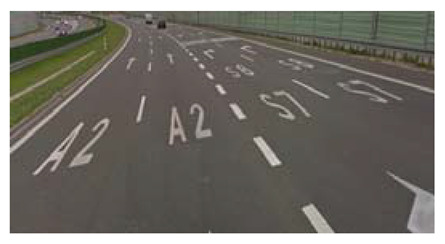	SR2 72%	42%	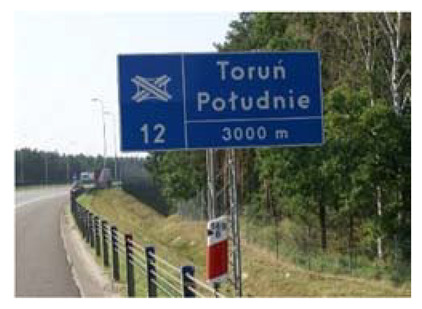	STR2 27%	29%
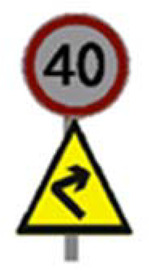	SR3 58%	4%	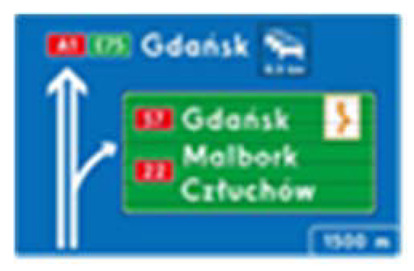	STR3 22%	7%
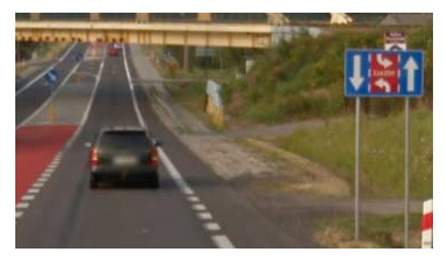	SR4 42%	13%	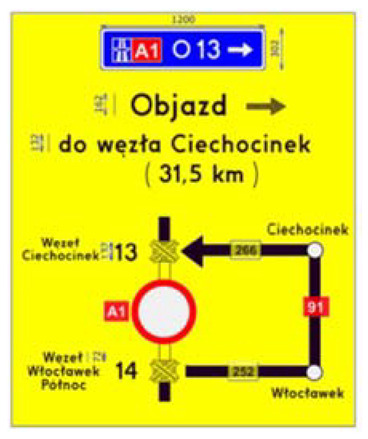	STR4 16%	32%
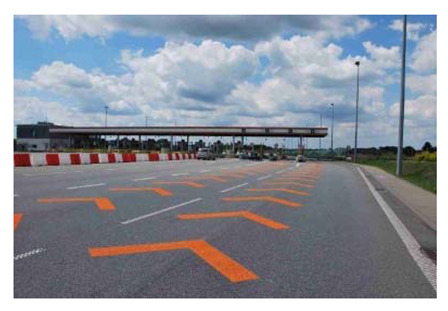	SR5 7%	26%	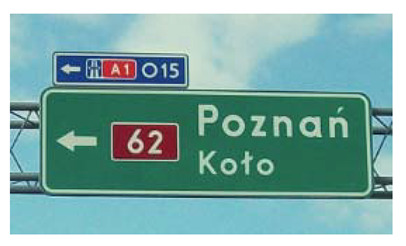	STR5 8%	18%
	46.6%	18%		24.7%	23.7
**Symbolic Warning Signs—SW**			**Symbolic and Text Warning Signs—STW**		
**Signs**	**Compehensibility**	**Familiarity**	**Signs**	**Comprehensibility**	**Familiarity**
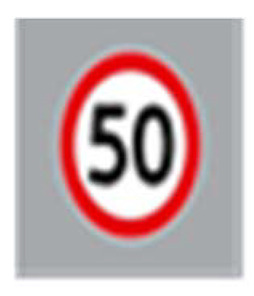	SW1 74%	66%	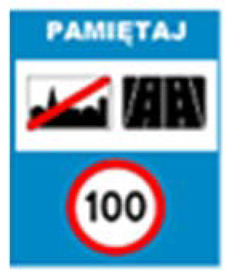 (remember about the speed limit)	STW1 17%	19%
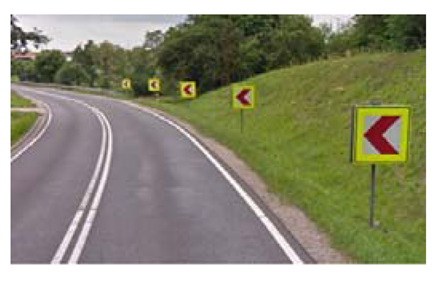	SW2 70%	73%	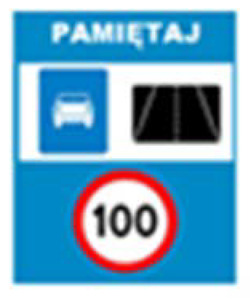	STW2 16%	19%
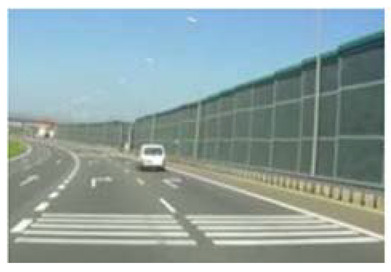	SW3 54%	8%	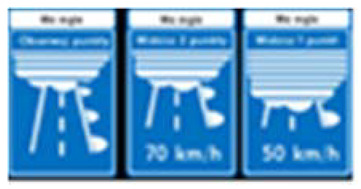 (driving in the fog)	STW3 32%	6%
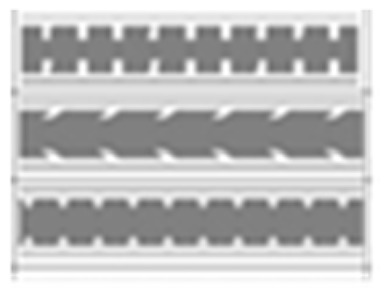	SW4. 41%	2%	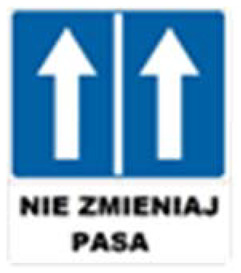 (don’t change lane)	STW4 8%	13%
	60%	37.3%		18.3%	14.3%

**Table 3 ijerph-18-02678-t003:** SPSS descriptive statistics of driver comprehensibility of traffic signs.

Traffic Sings (Comprehensibility)	N	Min	Max	Mean	SD
SR	369	0.00	5.00	2.37	1.0739
STR	369	0.00	5.00	1.57	1.1292
SW	369	0.00	4.00	2.45	0.9606
STW	369	0.00	3.00	0.74	0.7281

**Table 4 ijerph-18-02678-t004:** Linear regression coefficients between the familiarity and comprehensibility of traffic signs.

	R^2^	R^2^—Corrected	Beta	df	N	F	*p*
SR	0.0001	−0.002	−0.02	1	367	0.196	0.658
STR	0.049	0.046	0.22	1	367	18.592	0.000
SW	0.034	0.032	0.12	1	367	8.349	0.046
STW	0.004	0.001	0.06	1	367	1.349	0.246
All Sign Groups in Total	0.013	0.012	0.02	1	367	4.423	0.054

**Table 5 ijerph-18-02678-t005:** SPSS independent sample t-test on the effect of driver gender on traffic signs comprehensibility.

Comprehensibility of Traffic Signs	Gender	N	Mean	SD	t	*p*
SR	Male Female	218 151	2.50 2.19	1.116 0.984	7.465	0.007
STR	Male Female	218 151	1.65 1.47	1.162 1.119	2.164	0.142
SW	Male Female	218 151	2.49 2.39	0.912 1.030	0.838	0.361
STW	Male Female	218 151	0.73 0.75	0.714 0.750	0.053	0.818

**Table 6 ijerph-18-02678-t006:** SPSS ANOVA test for the relation between drivers’ age, driving experience, educational level, driving licence and familiarity and comprehensibility of traffic signs.

		Driver’s Age		Driving Experience		Driver’s Educational Level		Driving Licence Category	
Comprehensibility and Familiarity of traffic signs	N	F	p	F	p	F	p	F	p
Comprehensibility of SR	369	3.951	0.047	1.734	0.112	1.033	0.377	0.291	0.884
Familiarity of SR	369	1.532	0.091	27.723	0.000	0.384	0.534	7.462	0.006
Comprehensibility of STR	369	2.022	0.074	0.536	0.781	2.768	0.041	2.855	0.023
Familiarity of STR	369	2.354	0.041	2.511	0.023	1.034	0.378	1.923	0.091
Comprehensibility of SW	369	1.589	0.162	1.564	0.156	1.126	0.338	0.524	0.757
Familiarity of SW	369	2.534	0.021	7.378	0.000	0.484	0.694	1.357	0.248
Comprehensibility of STW	369	0.973	0.433	1.196	0.307	0.107	0.955	2.068	0.068
Familiarity of STW	369	1.611	0.153	0.798	0.571	0.737	0.531	3.595	0.007

**Table 7 ijerph-18-02678-t007:** Statistical measures of clusters between variables of personality traits and comprehensibility of nonstandard signs.

		Mean			SD			Statistical Significance of Differences Between Means	
		Cluster I	Cluster II	Cluster III	Cluster I	Cluster II	Cluster III	F	*p*
Comprehensibility of	SR	−0.598	−0.313	0.669	0.941	0.895	0.727	72.947	0.000
	STR	−0.169	0.113		0.902	1.044		6.495	0.000
	SW	−0.230	0.127		1.006	0.965		9.906	0.002
	STW	0.015	−0.001		1.016	0.997		0.024	0.876
Extraversion (EX)	SR	0.737	−0.676	0.217	0.921	0.866	0.7433	82.188	0.000
	STR	0.573	−0.323		0.968	0.864		79.059	0.000
	SW	0.557	−0.289		0.915	0.905		64.479	0.000
	STW	0.531	−0.298		0.969	0.879		66.088	0.000
Agreeableness (AG)	SR	0.851	−0.337	−0.221	1.018	0.783	0.878	54.671	0.000
	STR	0.802	−0.448		0.901	0.731		198.003	0.000
	SW	0.833	−0.452		0.903	0.744		189.625	0.000
	STW	0.919	−0.474		0.845	0.708		269.729	0.000
Openness (OP)	SR	0.624	−0.300	−0.101	0.949	0.988	0.879	26.883	0.000
	STR	0.491	−0.276		0.979	0.903		54.494	0.000
	SW	0.421	−0.212		1.005	0.924		32.782	0.000
	STW	0.405	−0.221		1.021	0.916		34.456	0.000
Neuroticism (NE)	SR	0.748	0.013	−0.504	0.959	0.831	0.897	52.642	0.000
	STR	0.373	−0.211		1.075	0.913		28.818	0.000
	SW	0.394	−0.244		1.066	0.931		31.454	0.000
	STW	0.499	−0.242		1.019	0.901		49.578	0.000
Conscientiousness (CO)	SR	0.660	−0.801	0.376	0.927	0.719	0.724	118.666	0.000
	STR	0.655	−0.352		0.835	0.884		108.405	0.000
	SW	0.672	−0.371		0.860	0.863		108.388	0.000
	STW	0.622	−0.331		0.866	0.886		94.082	0.000

## Data Availability

Data available in the project report DZP/RID-I-36/5/2016 “Experimental road marking in terms of behaviour of road users”.

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
