# Peer review of "Study on the Relationship between Drivers’ Personal Characters and Non-Standard Traffic Signs Comprehensibility"

_ijerph, 2021, doi:10.3390/ijerph18052678_

Round 1

Reviewer 1 Report

The paper summarizes research on the effectiveness of non-standard traffic signs to provide additional information to transport users and how to increase their comprehension and the impact.

After a good Introduction and Literature Review, the paper presents an unsatisfactory Research Methodology with very poor information about the experimental phase, the traffic signs selection criteria, the samples dimension, and expected reliability.

A general and crucial observation is the number of the 361 sample, which is very limited for the statistical analysis and the article's purposes.  Authors should better describe how the sample was selected and what would have been an adequate sample size representative for each group, highlighting the limitations of the analysis and conclusions.

It is not clear if the survey and questionnaires are collected after a driver ride on the road or showing a traffic sign to the drivers in controlled situation.

The Authors report not real and not an operative condition, i.e., driver's interview timeless to respond and the possibility to correct the answer after the first choice. In real conditions, the drivers have only a few seconds to respond to signs information, and it is tough to modify the behavior.

For this reason, the personality traits can have a high impact on collecting data in real conditions concerning the situation described by the researchers. For example, a neurotic driver can have a better non-standard traffic signs comprehensibility if it has a long time to reflect.

For this reason, the discussion of research results is not well supported by data and statistical analysis

Finally, the conclusion is generic and not specific since not report specific findings of the research

Particular remarks:

Fig.1 a (figure in the centre), and Fig.1 c (figure on the left) needs to be improved as well in Table 2.

In Table 1 the often the sum of each group characteristics is different from 369

In Tables 4 e 5 the meaning of simbols (R, d, f, N,  p, t) are not explained

Author Response

Rev: The paper summarizes research on the effectiveness of non-standard traffic signs to provide additional information to transport users and how to increase their comprehension and the impact.

After a good Introduction and Literature Review, the paper presents an unsatisfactory Research Methodology with very poor information about the experimental phase, the traffic signs selection criteria, the samples dimension, and expected reliability.

A general and crucial observation is the number of the 361 sample, which is very limited for the statistical analysis and the article's purposes. Authors should better describe how the sample was selected and what would have been an adequate sample size representative for each group, highlighting the limitations of the analysis and conclusions.

A: Thank you for a very valuable comment. People for the study were randomly selected based on the database of companies employing professional drivers. The entire fragment of the paper, describing the selection of the study group, has been changed and more details have been added. A subsection “Discussion of research results” has been also introduced, in which we drew attention to these limitations.

Rev: It is not clear if the survey and questionnaires are collected after a driver ride on the road or showing a traffic sign to the drivers in controlled situation.  

A: This is also a very valuable comment, for which we are thankful, and which has been included in the reviewed paper. We described in detail: how the study tools were developed, and on the basis of which criteria the results were analyzed.

Rev: The Authors report not real and not an operative condition, i.e., driver's interview timeless to respond and the possibility to correct the answer after the first choice. In real conditions, the drivers have only a few seconds to respond to signs information, and it is tough to modify the behavior.

A: Thank you for a very valuable comment. We have made changes to the paper by adding the description in detail of the study procedure.

Rev: For this reason, the personality traits can have a high impact on collecting data in real conditions concerning the situation described by the researchers. For example, a neurotic driver can have a better non-standard traffic signs comprehensibility if it has a long time to reflect.

A: We fully agree with this comment and we thank you for raising it. Indeed, when reading the paper, such an impression can be made. Therefore, we have complemented this entire fragment of the paper introducing changes to the text.

Rev: For this reason, the discussion of research results is not well supported by data and statistical analysis.

A: Thank you for this comment. We have once again carefully reviewed the entire text, which includes the analysis of results and discussions. All mistakes have been corrected by making changes to the text.

Rev: Finally, the conclusion is generic and not specific since not report specific findings of the research.

A: Thank you for this comment. The discussion and conclusions have been also significantly expanded to include the results on the drivers’ characteristics. The fragment of the paper concerning the conclusions has been also complemented. As a consequence, the literature has been also expanded.

Rev: Particular remarks:

Fig.1 a (figure in the centre), and Fig.1 c (figure on the left) needs to be improved as well in Table 2.

In Table 1 the often the sum of each group characteristics is different from 369

In Tables 4 e 5 the meaning of simbols (R, d, f, N, p, t) are not explained

A: We thank you also for the particular remarks. All of them have been included in the paper. The symbols used in the text are commonly used in statistical analyzes

Additionally, we attach the text with the changes - the corrections were made in the track changes mode

Reviewer 2 Report

Dear Authors,

very interesting article. Below are some observations.

  1. Line 45 - paragraph indentation is unnecessary.
  2. Line 151 - four photos counting from the right site are illegible.
  3. Lines 153 and 155 - as line 151.
  4. Line 196 - names of provinces should be write with a lowercase letter.
  5. Table 2 - some of signs are illegible.

Author Response

Thank you for this suggestion. We have changed the photos and the text

Reviewer 3 Report

This paper studies the level of comprehensibility of four different types of non-standard traffic signs: symbolic-regulatory, symbolic-text-regulatory, symbolic-warning, symbolic-text-warning. The research analyzes personality traits' influence on the comprehensibility and familiarity of the different non-standard traffic sign groups. It also explores the relationships between comprehensibility and familiarity. The study is based on questionnaires responded by 369 drivers in Poland. Statistical analysis of the results is based on regression analysis and ANOVA.

My overall impression is that the paper is relevant to the journal. The topic covered is motivating, and the research is well conducted. From a statistical point of view, the results are convincing, though many of the results are somewhat expected. Originality would be on the average/low level. Though the paper can be easily followed, its presentation must be improved to be published. Also, English should be carefully reviewed. As a general recommendation, I suggest the authors shorten long sentences. 

In the following, I provide detailed comments on the paper.

1) In the abstract, the Five-Factor Inventory is mentioned but not defined. I suggest the authors use the term "personality trait" in the abstract.

2) ln: 29-30. "The mechanism of traffic ... and effects". Please review this sentence. I do not understand its construction.

3) ln: 69-72. This sentence is unreadable.

4) ln: 135-136. Please review this point because it does not make sense to relate the level of comprehensibility with the level of comprehensibility.

5) I suggest defining the difference between regulatory and warning signs.

6) ln: 150-158. Some of the pictures do not have enough resolution. Also, try to group the figures more elegantly.

7) ln: 172-185. I understand the questionnaire used to obtain the respondents' descriptive data, familiarity, and comprehensibility. However, I do not fully understand how the responses are used in the statistical analysis for the familiarity and comprehensibility of data. I understand that success refers only to the option with the full and correct meaning for the comprehensibility statistic. However, It is not clear what you consider success in familiarity. Is it response number 5 on the familiarity scale?

8) The acronyms used to define the four groups of signals are not consistent. If you use SR and STR, you should use SW and STW instead of adding an S to SW and STW.

9) Table 1 should be more compact.

10) Table 2 is a mess. The headers are wrong. The results that show the table are not defined in the Table caption. Concerning my question 8), what do these percentages show? Comprehensibility seems to be the ratio of respondents whose response is exact. What about familiarity? Please, this should be clearly stated. Also, revise the spelling.

11) Tables 3,4,5,6 must be compacted. Use acronyms in the first column. The mean values are not defined. What do they show? What is the meaning of mean comprehensibility? This is very important.

12) ln: 268-269. This paragraph should be removed.

13) ln: 286. 2.511 p<0.041. Revise these values. They seem incorrect, according to the table.

14) ln: 309. p<9.023 is incorrect. It should be 0.023.

15) The idea of applying clustering to obtain personality profiles is very intuitive and interesting. However, the presentation of section 4.3 should be more compact. My recommendation is to show all the charts together in a figure with four subplots. Also, join the four tables in only one table. It should be explained in the caption and the text the meaning of the y values regarding each chart. It is clear from the charts that the vertical axis shows the means of each factor in the clusters. However, what about the comprehensibility scale? Finally, why Figure 4 is in black and white?

16) In the discussions: ln: 388-389, revise "The level of comprehensibility... this study". I cannot understand its meaning.

Author Response

A: Thank you for all your comments and suggestions. All of them have been included in the reviewed paper.

Rev: This paper studies the level of comprehensibility of four different types of non-standard traffic signs: symbolic-regulatory, symbolic-text-regulatory, symbolic-warning, symbolic-text-warning. The research analyzes personality traits' influence on the comprehensibility and familiarity of the different non-standard traffic sign groups. It also explores the relationships between comprehensibility and familiarity. The study is based on questionnaires responded by 369 drivers in Poland. Statistical analysis of the results is based on regression analysis and ANOVA.

My overall impression is that the paper is relevant to the journal. The topic covered is motivating, and the research is well conducted. From a statistical point of view, the results are convincing, though many of the results are somewhat expected. Originality would be on the average/low level. Though the paper can be easily followed, its presentation must be improved to be published. Also, English should be carefully reviewed. As a general recommendation, I suggest the authors shorten long sentences.

In the following, I provide detailed comments on the paper.

1) In the abstract, the Five-Factor Inventory is mentioned but not defined. I suggest the authors use the term "personality trait" in the abstract.  

A: Thank you for your comment, which we have included it in the text. We also wanted to add that in the discipline of psychology the concept of personality traits is commonly used.

Rev: 2) ln: 29-30. "The mechanism of traffic ... and effects". Please review this sentence. I do not understand its construction.

A: Thank you for your comment. This fragment has been changed.

Rev: 3) ln: 69-72. This sentence is unreadable. 

A: Thank you for your comment. We have changed the text to make this sentence readable.

Rev: 4) ln: 135-136. Please review this point because it does not make sense to relate the level of comprehensibility with the level of comprehensibility.  

A: Thank you for this comment. We have changed the text to make this sentence readable.

Rev: 5) I suggest defining the difference between regulatory and warning signs.

A: Thank you for this comment. We have added the description of the differences between regulatory and warning signs.

Rev: 6) ln: 150-158. Some of the pictures do not have enough resolution. Also, try to group the figures more elegantly.

A: Thank you for the suggestion. The pictures have been corrected.

Rev: 7) ln: 172-185. I understand the questionnaire used to obtain the respondents' descriptive data, familiarity, and comprehensibility. However, I do not fully understand how the responses are used in the statistical analysis for the familiarity and comprehensibility of data. I understand that success refers only to the option with the full and correct meaning for the comprehensibility statistic. However, It is not clear what you consider success in familiarity. Is it response number 5 on the familiarity scale?  

A: Thank you for a very valuable comment. We have made changes to this fragment of the paper by adding more details.

Rev: 8) The acronyms used to define the four groups of signals are not consistent. If you use SR and STR, you should use SW and STW instead of adding an S to SW and STW

A: Thank you for this valuable comment. We have changed the text to include your suggestions.

Rev: 9) Table 1 should be more compact.

A: Thank you for your suggestion. The table has been compacted.

Rev: 10) Table 2 is a mess. The headers are wrong. The results that show the table are not defined in the Table caption. Concerning my question 8), what do these percentages show? Comprehensibility seems to be the ratio of respondents whose response is exact. What about familiarity? Please, this should be clearly stated. Also, revise the spelling.

A: Thank you for your comment.The table has been corrected.

Rev: 11) Tables 3,4,5,6 must be compacted. Use acronyms in the first column. The mean values are not defined. What do they show? What is the meaning of mean comprehensibility? This is very important.

A: Thank you for your suggestion. The tables have been corrected.

Rev: 12) ln: 268-269. This paragraph should be removed.

A: Thank you for this comment. This fragment of the paper has been corrected.

Rev: 13) ln: 286. 2.511 p<0.041. Revise these values. They seem incorrect, according to the table.

A: Thank you for this comment. The mistake has been corrected.

Rev: 14) ln: 309. p<9.023 is incorrect. It should be 0.023.

A: Thank you for this comment. The mistake has been corrected.

Rev: 15) The idea of applying clustering to obtain personality profiles is very intuitive and interesting. However, the presentation of section 4.3 should be more compact. My recommendation is to show all the charts together in a figure with four subplots. Also, join the four tables in only one table. It should be explained in the caption and the text the meaning of the y values regarding each chart. It is clear from the charts that the vertical axis shows the means of each factor in the clusters. However, what about the comprehensibility scale? Finally, why Figure 4 is in black and white?

A: Thank you for this suggestion. We have changed the text including tables and figures.

Rev: 16) In the discussions: ln: 388-389, revise "The level of comprehensibility... this study". I cannot understand its meaning.

A: Thank you for this comment. We have changed the text to make this sentence more understandable.

Additionally, we attach the text with the changes - the corrections were made in the track changes mode

Round 2

Reviewer 1 Report

The paper has been satisfactorily improved in most of the comments

However, the new version of the paper not overcomes the inherent limitations in the data collection, in particular the driver's interview timeless to respond or the opportunity to change the first response.

The Authors have to point out this limitation in the paper.

Author Response

Rev.:

However, the new version of the paper not overcomes the inherent limitations in the data collection, in particular the driver's interview timeless to respond or the opportunity to change the first response.

The Authors have to point out this limitation in the paper.

A: The text has been supplemented with the following note:

"One should remember that the reaction study was not carried out in realistic conditions. Because of this, responders had significantly longer time for reaction compared to real situations. Therefore, more significant attention has been paid to the assessment of  analysis differences in each separated groups than to the level of each signs understanding.

As a future work, tests in a driving simulator are planned, which should eliminate the specified limitations."

Reviewer 3 Report

First of all, I would like to thank the authors for providing answers to my comments. The manuscript has been significantly improved and I believe it is now easier to read and follow. In addition, almost all my concerns have been clearly answered and addressed when necessary. The tables and figures are now more compact and easy to read. I believe the manuscript would now be ready to be accepted after several corrections.

1) This is the most important point. I still cannot see what "Mean" represents in the different Tables. For instance, in Table 3, what is the meaning of 2.3739? This is supposed to be the mean comprehensibility value. However, you do not explain which is the nature of this value. In Table 2, it is now clear that comprehensibility and familiarity are measured as a percentage. However, the "level" of comprehensibility is not defined anywhere.

2) In 71-72 I suggest changing the sentence to: "The application of signals, whose understanding depends largely on the personal traits of the driver, should be avoided."

3) In Table 2, second row: please correct "compehensibility".

4) After reviewing Figure 2 again, which I believe now is more compact and easy to read, I would do the following. Remove the SR, STR, SW, and STW ticks and the corresponding line segments. Change the vertical axis caption "comprehensibility" to "Personal Profile Values". Finally, use a legend beside each line showing a cluster to put the comprehensibility value associated with each cluster (for instance with the same font color used to plot each line). Mixing values of different parameter types in the same line is confusing and unusual.

Congratulations for your work.

Author Response

Rev.:

1) This is the most important point. I still cannot see what "Mean" represents in the different Tables. For instance, in Table 3, what is the meaning of 2.3739? This is supposed to be the mean comprehensibility value. However, you do not explain which is the nature of this value. In Table 2, it is now clear that comprehensibility and familiarity are measured as a percentage. However, the "level" of comprehensibility is not defined anywhere.

A: The text has been supplemented with the following note:

"In order to estimate statistical significance of differences understanding when analysing 4 separate groups of non-standard signs, the mean values of the correctness of understanding the meaning of a given group of signs by each of the respondents have been found. 1 point was given for a correct recognition of each sign (and not a group of signs). If the recognition was incorrect, then a number 0 was assigned. So, the minimal number of points was 0 (in the case when none of signs was correctly described), while the maximum was: 5, 4 and 3 for SR/STR, SW and STW signs, respectively. Having the information about the mean values for each group of signs (369 checked respondents), the statistical significance of differences between them was possible to assess."

Rev.: 2) In 71-72 I suggest changing the sentence to: "The application of signals, whose understanding depends largely on the personal traits of the driver, should be avoided."

A.: This paragraph has been changed

Rev.: 3) In Table 2, second row: please correct "compehensibility"

A.: A correction has been made

Rev.: 4) After reviewing Figure 2 again, which I believe now is more compact and easy to read, I would do the following. Remove the SR, STR, SW, and STW ticks and the corresponding line segments. Change the vertical axis caption "comprehensibility" to "Personal Profile Values". Finally, use a legend beside each line showing a cluster to put the comprehensibility value associated with each cluster (for instance with the same font color used to plot each line). Mixing values of different parameter types in the same line is confusing and unusual.

A.: A correction was made in accordance with the reviewer's indications